# Participatory Approaches to Understand Dietary Behaviours of Adolescents in the Secondary School Setting

**DOI:** 10.3390/nu12123761

**Published:** 2020-12-07

**Authors:** Sarah Browne, Carol Barron, Anthony Staines, Mary Rose Sweeney

**Affiliations:** 1School of Public Health, Physiotherapy & Sports Science, University College Dublin, Dublin 4, Ireland; 2School of Nursing, Psychotherapy and Community Health, Dublin City University, Dublin 9, Ireland; carol.barron@dcu.ie (C.B.); anthony.staines@dcu.ie (A.S.); maryrose.sweeney@dcu.ie (M.R.S.)

**Keywords:** adolescent health, participatory research, peer led research, photovoice, focus groups, school food, dietary behaviours, food choice, food environments

## Abstract

Acknowledgement of wider socio-ecological factors that influence dietary behaviours needs greater consideration in nutrition research with young people. Additionally, children and adolescents have a right to have their voices heard in research that concerns them. The aim of this methods paper is to describe and evaluate participatory methodologies undertaken as part of a dietary behaviour study with adolescents in the school setting in Ireland. Photovoice and peer-led focus groups were the key participatory methodologies, undertaken alongside food diaries and anthropometry. These methodologies were evaluated through discussion with peer researchers, qualitative surveys and in the context of the wider study process and outcomes. Peer researchers reported learning new skills including research, facilitation, listening and social skills and many felt they gained confidence, as well as an awareness about healthy and unhealthy practices at school. The findings were found to be authentic according to students, except for two limitations. Students believed body image was not adequately represented in the findings, and alternative focus group compositions could have influenced discussion content. Youth participants were afforded genuine opportunities to have their voices heard as part of a diet and nutrition research and the methodologies were acceptable and enjoyable. They demonstrated agency in valuable contributions at project design, data collection, analysis and interpretation stages of the research process. Furthermore, the participatory methodologies complemented quantitative data by providing environmental, behavioural, and socio-cultural insights into food choice in the school setting.

## 1. Introduction

The adolescent years represent a time of rapid growth and development, with high nutritional needs. Dietary patterns, however, disimprove compared to childhood, as access, social routines, and autonomy around food choice change [1,2]. Overweight and obesity in this age group is an immediate global concern, with socio-environmental factors linked to the rapid increase in prevalence in recent decades [3,4]. Everyday settings like schools, therefore, are important spaces for health promotion initiatives with young people and the sector has responsibilities in providing healthy eating education and food environments that can support healthy choices [3,5].

Adolescents source more free sugars, saturated fats, and sodium outside the home compared to inside [6]. Schools can be a source of low quality foods and beverages; however, local food outlets also need to be incorporated into our current concept of the school food setting [1,7,8,9,10]. A common thread in the literature is the influence of fast food within 1 km of schools, associated with greater consumption of low nutrient, energy dense foods and less high quality foods such as fruits and vegetables [9,11,12]. Competition with local food outlets, therefore, is a challenge to nutrition policies in secondary or high schools [1,7,8,9,10,11,12]. Additional considerations that school food reform can sometimes ignore are individual, interpersonal, social and temporal factors that are important to students [1,7,8,13,14,15]. Wider socio-ecological factors, therefore, need ongoing consideration in nutrition research that intends to understand and influence adolescent behaviour in the school setting.

In this study, a rights-based approach to research with young people was taken. The United Nations Convention on the Rights of the Child (UNCRC) outlines how children should be treated and that children and young people are ‘rights holders’ [16]. By ratifying the UNCRC in 1992, Ireland made a commitment under international law to respect, protect and fulfil children’s rights as they are set out in the Convention. Article 12 of the UNCRC is important in relation to the rights of young people to have their voices heard about matters that concern them:
“States Parties shall assure to the child who is capable of forming his or her own views the right to express those views freely in all matters affecting the child, the views of the child being given due weight in accordance with the age and maturity of the child”.Article 12 of the UNCRC, 1989 [16]

The rights-based approach in this study follows the work of Lundy amongst others [17]. The methodology focuses on research with, rather than research about, children and young people. Participatory methodologies have contributed to research that can better influence practices and policies which are youth-centred and appropriate to adolescent’s contemporary circumstances. There is a growing body of literature that incorporates participatory approaches with young people in food and nutrition research [14,18,19,20,21]. A number of researchers report on the importance of engaging sensitively and reflexively, which can lead to reduced power imbalances in the research process and can provide insight and understanding into experiences that are not easily conveyed or captured using traditional methodologies [22,23,24,25]. However, there is minimal literature which evaluates differing research techniques with young people by young people.

This methodology paper describes a study with an adult-led research question in which adolescents were invited to take up meaningful roles. Lundy’s conceptualization of Article 12 of the UN Convention on the Rights of the Child was designed specifically for an examination of children’s participation and focuses on four components of space, voice, audience and influence [17]. The application to this study means that: (1) safe and inclusive opportunities were provided to adolescents to form and express their views, (2) adolescents were facilitated in expressing their views, (3) adolescents were listened to and (4) their views were acted upon.

The aim of this paper is to describe participatory methodologies undertaken as part of a larger dietary behaviour study with adolescents in the school setting in Ireland, to describe participant evaluations of the methodologies, and to share insights on the process in the context of dietary behaviour research.

## 2. Materials and Methods

### 2.1. Study Design

A cross-sectional observation study with a sample of six Irish secondary schools was conducted. The aim of the study was to understand the socio-ecological factors influencing food choice for adolescents in this setting. Students, teachers and principals took part in qualitative methodologies [1], and students also participated in quantitative methodologies with regard to dietary behaviour and anthropometry [6]. In line with a rights-based approach to research with young people, a sub-sample of students participated as ‘peer researchers’, which involved moderating focus groups with peers and conducting observations of the school and local food environments through photovoice. Peer researchers also took part in debrief and feedback sessions with the adult researcher.

### 2.2. School Recruitment

Active student involvement in the research study was a key piece of information provided by the researchers to principals and link teachers when recruiting schools. Public secondary schools with a transition year programme were eligible because dedicated curriculum time for social and community projects is provided during the programme, and this aligns well with a participatory research approach. Additionally, to capture local food environments students would need permission to leave school grounds at lunch time, and this permission is typically in place by transition year. School characteristics and participants are outlined in Table 1.

### 2.3. Student Recruitment

The various aspects of the study were comprehensively explained to students in an assembly. Students were provided with information leaflets for themselves and their parents that outlined the participatory processes and other aspects of the study clearly. Students were given the option on their consent and assent form to opt-in for some or all aspects of the study, including as food diary and anthropometry participants, peer researchers (focus group moderators and photovoice researchers) and focus group participants. Where there were more volunteers than required for peer-led data collection, peer researchers were randomly selected using a random numbers table. Students returned parental consent and individual assent forms to the liaison teacher. Table 1 describes school characteristics and student participation in the different strands of the study.

### 2.4. Quantitative Methodologies

#### 2.4.1. Food Diaries

The rationale for conducting food diaries was to capture quantitative information on food and nutrient consumption and sources of food at school with the objective of comparing nutrition quality of food intake from three sources—home packed lunches, school sourced lunches and lunches sourced in the local retail environment. Methods and results have been published elsewhere [6]. To briefly summarise, participants recorded the time of day, eating location, foods and beverages consumed, portions sizes of items consumed and food source over four days.

#### 2.4.2. Anthropometry

Anthropometric measures were used to classify the sample according to weight, height, BMI and waist circumference for age. Additionally, direct measures of body weight were used to assess the accuracy of food records using methodologies outlined by Black [26]. Trained researchers with healthcare backgrounds (physicians, nurses, dietitian) conducted the weight, height and waist circumference measurements adhering to best practice [27]. The anthropometric protocols were evaluated with students via semi-quantitative self-report survey with students indicating either a ‘positive’, ‘negative’ or ‘neither positive or negative’ experience and providing reasons for their choice in an open comment box. The responses were analysed for frequencies and open comments analysed for content and categorised into themes.

### 2.5. Development of Participatory Methodologies for the School Setting

Two key participatory methodologies will be the focus here: (1) Photovoice and (2) Peer led focus groups. A key step in developing the methodologies included preplanning consultation with secondary schools and students, and thereafter piloting peer-led focus groups and photovoice. In line with recommended practice, a youth advisory panel was consulted from the design stage [28]. Youth panel members (aged 15–17 years) represented urban and rural schools and met with the researcher on three occasions. The terms of reference included consultations on study design, procedures, and language used for tools and resources. They advised on the choice of recruitment process for “peer researchers”, preferring volunteering to teacher/student nominations. They suggested incentives for peer researchers and online music vouchers were recommended as an acceptable compensation for their time and effort. Their review of the language and phrasing of questions for use in written material and focus group questions was valuable in implementing acceptable language. Furthermore, a degree of flexibility was incorporated into the study design so that participants could later influence data collection in their respective schools—e.g., adding new focus group questions or suggesting alternative settings for photovoice data collection. All methodologies were interactive, participatory, creative, and youth-friendly while being focused on the study aim.

### 2.6. Survey Instruments/Qualitative Approaches

#### 2.6.1. Photovoice

The use of image-based data originates in the fields of anthropological and ethnographic studies [29] and has transferred to childhood studies in various disciplines. Photography is a part of most adolescents’ daily lives through social networking and, therefore, is a methodology that they are very familiar with. Our approach was informed by previous work [23,30,31], and was further adapted for local school setting factors informed by participants in the youth advisory panel and pilot studies.

Peer researchers attended a short workshop for 1½ hours at school, at which they were instructed on using the disposable cameras provided, and on annotating photographs. Information was provided about ethical issues, including respecting the privacy of students and staff (i.e., avoid identifiable imagery) and being transparent about their activity with peers. Peer researchers were informed that food availability, food pricing, food service and dining infrastructures, food marketing, nutrition messaging, and relevant student behaviours in both the school and local food settings were relevant to the study. They were asked to record additional descriptive notes about their school and local food environments on a form provided.

Peer researchers, working in pairs, were provided with blank observation forms, clip-boards, pens and disposable cameras and arrived at school early. They had permission to miss classes over one day to complete the research and continue observations through break and lunch times. At the end of the school day, the researcher met with the students to discuss their experience. Notes of their feedback were written down, and peer researchers filled in a qualitative evaluation questionnaire.

Visual content analysis following Bell [32] was used to analyse photographs and the process is described in more detail elsewhere [33]. Bell describes visual content analysis as an “*empirical and objective procedure for quantitatively recording visual representations using reliable and explicitly defined categories*” [32]. Content in photographs was supplemented by peer researchers’ annotations, fieldwork notes and audio-records of debrief discussions. Feedback sessions were held with peer researchers to discuss their views of the dominant themes.

#### 2.6.2. Peer Led Focus Groups

Peer led focus groups, where adolescents become the moderators, are based on a premise of minimizing adult influence within shared social networks and the method attempts to address power imbalances inherent in youth focused research [34]. A strong rationale for community involvement is that, in certain groups, an outside researcher may not be able to elicit in-depth, inside data that a peer or community member could [35]. This is relevant for adolescents, who are in a transition stage between parental/adult control and independence. However, there are few published studies evaluating this method among children or adolescents. We were guided by existing work on peer-led focus groups by Murray [34], who explored offending behaviour among adolescents. We sought to evaluate how this methodology could be adapted to dietary behaviour research.

A training workshop for students who volunteered to lead focus groups was designed for this study. The content of the training was informed by literature on traditional focus group moderation [35,36] and participatory methodologies [34,37,38], as well as input from the youth advisory panel described earlier.

Six to eight student volunteers attended a 3 h training workshop at each school with the researcher (SB). The workshops were conducted in a focus group style whereby chairs were set up in a circle and the session was audio-recorded. The content of the workshop included how to set ‘group agreements’ at the outset, learning about open and closed questions, probing for greater discussion, tools on dealing with difficult behaviours in groups, and time for skills practice. After the training students decided who would go on to moderate the focus group, who would assist, and who would prefer neither role. Some students opted to share moderation so that they could support each other. Extra students attended the training workshop to allow for drop-outs and absenteeism. Students were responsible for running focus groups in the school setting, which were audio-recorded for later transcription and analysis.

Greater detail of the analysis is described elsewhere [1]. In brief, conventional content analysis [39] was used to analyse the focus groups through a socio-ecological lens. Peer researchers listened to their respective recording and highlighted/noted important dialogue on a hard-copy of the transcript, which was accounted for in the analysis conducted by two researchers (SB, CB). Feedback sessions were held with students to discuss their views of the dominant themes.

### 2.7. Ethics

The project was granted full ethical approval from the research ethics committee at Dublin City University (DCUREC/2012/114). All students provided informed parental consent and individual assent before participating in the study.

## 3. Results

### 3.1. Student Evaluation of Methodologies

All peer researchers reported learning new skills that included research, facilitation, listening and social skills and many felt they gained confidence through the process. Matthew, for example valued “learning when to listen more and take in information people feel strongly about”. Awareness-raising was appreciated and widely reported whereby students learned more about their school and the behaviours of peers. It made them take notice of healthy and unhealthy practices and some students reflected on their own behaviours as a result: “I realise I should eat healthy and exercise more after my experience” (Lisa). The ‘fun’ aspect was also important, and students enjoyed taking time from the usual school routine to be involved in a novel project.

Students moderating focus groups thought that they ‘worked’ and, for them, small group discussions without the presence of an adult allowed people to open up comfortably. They viewed the discussions as accurate because they could relate to what they were hearing from peers, as reflected in Sophie’s comment: “They opened up and confessed some thoughts to us and maybe more than usual because we’re all equal”.

### 3.2. Limitations Identified by Students

In training, peer moderators were advised not to ask leading questions or share their own opinions as this might give the impression there were ‘right’ answers, particularly since peers would be aware that moderators had attended training. Some found it difficult not to do this and talked this through in debrief sessions. In the recordings, some examples were found where leading questions, attempts to build full consensus, assumptions or ill-timed interruptions seemed to limit discussion flow. Peer moderators did find silences or lulls in conversations a challenge. However, more examples were obvious where peer moderators used their insider knowledge in a natural way to advance or deepen discussions. Additionally, at the transcription review stage, peer moderators highlighted discussion points that concurred closely with the overall dominant themes being conducted in analysis by the researchers with focus group recordings across all schools. These latter two points strengthen the argument for providing flexibility in participatory research and offering young people opportunities to influence the research process at all stages, including data collection and analysis.

Student feedback on ways to improve the research process showed critical evaluation skills. Focus group moderators advised that more diverse discussion would be possible by changing group composition. In mixed gender schools some felt that single gender groups would have elicited a better discussion. Some groups generated very little conversation about physical activity and sport, which they acknowledged would be important, and their explanation was the absence of ‘sporty’ students within the group. They felt a mix of ‘sporty’ and ‘non-sporty’ students would have generated better conversation on how this individual-level identity would influence dietary behaviour. The students undertaking photovoice observations suggested carrying out the data collection for longer than one day, setting up agreements with managers from local food retailers to capture this setting more comprehensively, and some found school canteen management suspicious of their activity.

When the researcher (SB) met students to share the study findings, feedback indicated they agreed with key findings, however also identifying an important gap. Body image did not feature as a theme or within any themes in results and students believed this was inaccurate given their lived experience. They believed that body image plays a big role in influencing dietary behaviours, even in the school setting. Healthy or clean eating and fitness had become very trendy for many in the previous one to two years. One student suggested: “maybe people were just embarrassed to say that they wanted to look well” when we explored why it might not have been discussed more during the focus groups. It is important to highlight, therefore, the limitations of participatory research in this instance and how other approaches were needed to explore this topic meaningfully.

### 3.3. Student Views on Participating in Anthropometry

The majority of students (88%, *n* = 268) who participated in anthropometry reported it as a positive experience and their reasons fit into one of three categories: (1) interest in finding out their measurements, (2) satisfaction with communication and study protocol (i.e., privacy), and (3) taking part was enjoyable. Comments from students who found it ‘neither positive or negative’ (18%, *n* = 55) indicated that either they had no strong reaction to being measured or satisfaction with how measurements were conducted, but dissatisfaction with their body. A minority of students (2%, *n* = 7) reported being unhappy with their weight or height and this was the reason they reported anthropometric participation as a negative experience.

## 4. Discussion

This paper describes an evaluation of participatory methodologies undertaken as part of a wider dietary behaviour study to understand socio-ecological influences on food choice of adolescents in the Irish secondary school setting. Peer-led focus groups and photovoice were enjoyed and acceptable to students, who reported new skills, insights, and awareness. Importantly, an authentic and accurate representation in findings was reported. The outcomes complemented a traditional quantitative food record and nutrient analysis approach by providing depth, context and understanding of environmental and behavioural factors that have relevance for future interventions. This discussion will explore the participatory approaches undertaken and the lessons learned.

### 4.1. Comparison with Prior Research

Participatory research literature emphasises that identifying problems is the first step in creating conditions for the generation of solutions and social action [17,30,40,41]. Our participatory work indicated that students could play a more active role in shaping their food experiences at school. However, genuine participation, where real opportunities for influence are available, can be challenging to fit into traditional education settings [40,42]. It is also still the case that physical spaces (e.g., school buildings, infrastructures, recreational spaces and so forth) designed for children and adolescents, what Rasmussen [43] terms “spaces for children”, largely remain adult designated and designed spaces, which still largely reflect adult values [44]. An earlier publication from this study, where students’ clear call for healthier food environments as a solution to improve dietary behaviours contrasted starkly with teachers and principals’ views on traditional education-based solutions, suggests that adults are more influential with regard to school food environments [1].

Peer led focus groups are not well described or evaluated in published literature. Exceptions include work on offending behaviour by Murray [34] and a recent study around moral and social values by Djohari and Higham [45] and in both there are similarities with some of our experiences. Skills training and a discussion framework seem to help guide the process but flexibility that allows peer moderators to follow group priorities is also important. Djohari and Higham incorporated a system of lead and assistant peer moderators [45], which worked well in our study too and helped peer researchers feel supported. Traditional focus group moderation is based on a level of detachment so that leading enquiries are avoided to minimize response bias. However, our work and others show that adolescent moderators have an alternative role whereby they need to be both facilitator and participant in order maintain a natural rapport with peers [34,45].

The photo-elicitation method is widely used in child and adolescent research [23,31,46] and has been used by some to describe food environments in and around schools [14,47]. Our findings align with other evaluations, whereby self-confidence and competence, critical awareness, and self-realisation or empowerment are important outcomes for participants in health-related research [30,48]. Wang et al. encourage us to evaluate the process holistically by categorising the potential advantages of photovoice for “participants with most power” and “participants with less power” [30]. Innovating in the school food context and learning from student’s expertise were key advantages for researchers, where student enquiries yielding contextual and nuanced findings added new understanding. As an example, one issue highlighted in photovoice, not identified through other methodologies, was location of free drinking water for students. Water fountains were captured in toilet facilities and taps on outside walls, with one in a refuse laden outdoor corner. None were in the dining areas of schools as recommended [49,50] and contrasted starkly with photographs of fridges stocked with brightly coloured sugar-sweetened beverage for sale. This raised awareness of conditions among students and led to critical discussion of the influence of environmental factors on behaviours. The “increase access to power” that Wang et al. [30] describe as a potential advantage for participants was evident when photovoice allowed students to convey meanings creatively and safely about their physical environments.

### 4.2. Practical Considerations for Translation

According to Lerner et al. youth participation is most likely to succeed when positive adult relationships are fostered, and skill building and opportunities for leadership in meaningful activities are offered [38]. Participant engagement at various stages, as well as workshops, training and research supports offered by the study, demonstrated a framework in which participatory research at school can happen effectively. Attention to practical aspects of communication pathways with schools is recommended after our experience. In this study, strong partnerships with link teachers greatly assisted in successfully navigating various study stages including parental and student consent, timetabling, and facilitating the use of school spaces for data collection. In terms of project planning, extra time was designated to consult with youth panels, run training workshops, and debrief and feedback sessions. Many of these activities are now accepted as integral patient and public involvement (PPI) recommended for research [29,51] and collaboration as a right for research participants is rapidly becoming a requirement in grant funding. There are a number of frameworks available to guide this process when working with young people and most agree on some core principles including listening to young people, supporting them to express their views, taking their views into account, involving young people in decision making processes and sharing responsibility and power in the decision making [52]. A key recommendation based on our evaluation would be to consult and collaborate with adolescents at as many stages as possible of the research process right up to interpretation and dissemination, in order to instill confidence that the authentic ‘voice’, opinions and experiences of participants are well captured.

A review of photovoice in diet and nutrition research published in 2010 found limited examples of the methodology in the field [48]. However, the use of the methodology is growing in the last 10 years, particularly with youth from minority communities [14,18,19,21], where it can more readily explore cultural, behavioural and environmental influences in a safe and accessible way. There are many examples of intervention studies that facilitate students to lead food changes in the school setting [40,53,54,55]. Published studies on problem identification and evaluation of study design are less well represented. Limited published studies on peer-led focus groups with adolescents were available when we designed our study and one paper guiding the methodology here was in the field of sociology and criminal behaviour [34]. The process translated well to dietary behaviour research, and represents a participatory methodology that can avoid tokenism, which is an inherent risk for researchers and youth participants [52,56].

### 4.3. Strengths and Limitations

A key strength is that principles of youth participatory research were upheld, and furthermore were authentic according to participants. The sample included adolescents from urban, rural and disadvantaged backgrounds; however, more diverse groups including ethnic minorities and students with intellectual or physical disabilities were not represented and further exploration of the methodologies with these groups is recommended. Students from a disadvantaged school opted not to participate in food diaries and anthropometry, adding a demographic limitation to quantitative outcomes. Indeed, known reporting errors in the food diary methodology [57] were evident in our sample, as well as a strong bias towards female representation compared to males [6]. The qualitative participatory methods, therefore, offered a real opportunity for students to use their voice as an alternative to traditional approaches. As a result, it is possible that the qualitative outcomes offer a more authentic representation of their dietary behaviours in this setting.

One limitation of photovoice is that issues raised are those that are easily photographed and that the participants value, rather than complete representations of peer perspectives in the same setting [47,58]. This limitation is applicable to peer-led focus groups too and, indeed, students identified it when consulted in feedback session in relation to sporty identities and body image influences, not well explored by the study. One potential consideration for addressing these limitations is that of recruitment strategies. Murray, for example, chose close friendship network recruitment for a peer-led approach, which may ensure that participants share a closer social world and address individual identities more closely [34]. Equally, however, we should not unquestionably assume that participatory research is more appropriate for conducting research with young people in all instances and it does not automatically make research ‘better’ [23]. Indeed, body image was a topic identified as challenging—both in relation to a safe space to discuss with peers and the importance of creating safe, communicative spaces when being measured as part of nutrition study.

Given the known body image sensitivities in the adolescent years, an alternative could have been to discuss this explicitly with young people earlier at the design stage. Body image was not named in the topic guide used by peer researchers, which included open questions on perceptions of health, how food choice is made, the influence of different settings, and sources of information on food and nutrition [1]. Furthermore, we did not observe references to body image by participants in audio-recordings so that no opportunities were obvious for peer researchers to pursue the topic further. It is possible that students inherently knew that body image was ‘off limits’ for discussion with peers in the space provided, which is interesting and needs interpretation as a possible outcome of this evaluation. Explicitly keeping topics off limits is one potential solution to avoiding sensitive topics suggested by Sim and Waterfield’s paper on ethical challenges in focus groups [59]. However, our evaluation indicates that an alternative was needed for students to express their views. To maintain authenticity in youth participatory research, there is an important role for experienced researchers to match methodologies appropriately, provide support and evaluate at various timepoints. Careful consideration is needed, therefore, to avoid disempowering the youth voice when participatory methodologies are not well matched to the topic or group [60].

## 5. Conclusions

Article 12 of the UNCRC has prompted the shift from conducting researching on children and young people to research with children and young people thereby enabling the young person’s voice to be heard more clearly and the methodologies described are faithful to this approach. We also acknowledge instances where alternatives to participatory methodologies need to be considered.

We propose that photovoice and peer led focus groups are valuable and effective methodologies to deepen our understanding of food choice and dietary behaviours. As adjuncts to a traditional quantitative approach, the participatory methodologies described and evaluated can access hard to reach information required for assessing healthy eating intervention needs in the school setting. Additionally, the importance of collaboration and co-design with young people cannot be underestimated so that we address their rights to participate, have their voices heard and have meaningful influence into how their spaces are shaped.

## Figures and Tables

**Table 1 nutrients-12-03761-t001:** Characteristics of participating secondary schools and students participating in each photovoice, focus group, food diary and anthropometry methodologies.

School	Gender	Eligible Students *n*	Peer Researchers Photovoice Participants *n* M/F	Peer Researchers Focus Group Moderators *n* M/F	Focus Group Participants *n* M/F	Food Diary Participants *n* M/F	Anthropometry Participants *n* M/F
School A	Urban Male only	55	2/0	2/0	8/0	21/0	1/0
School B	Urban Female only	116	0/2	0/2	0/8	0/85	0/1
School C	Rural Male only	46	2/0	3/0 ^∞^	7/0	37/0	0/1
School D	Rural Female only	78	0/2	0/2	0/6	0/60	0/1
School E	Urban Male & female	209	2/2 **	2/3 **^,∞^	10/9	34/68	0/1
School F	Urban DEIS * Male & female	80	0/2	½ ^∞^	2/6	0/0	0/0
Total		584	6/8	8/9	27/27	92/213	1/4

* Delivering Equality of Opportunity in Schools (DEIS)—Disadvantaged Status; ** Two focus and photovoice groups were conducted owing to larger student body in school; ^∞^ Three peer focus group moderators were preferred by three groups—two shared the moderator role, while one assisted with note taking. All participants were between 15.2 and 17.3 years at the time of data collection.

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
