# Peer review of "Participatory Approaches to Understand Dietary Behaviours of Adolescents in the Secondary School Setting"

_nutrients, 2020, doi:10.3390/nu12123761_

Round 1
Reviewer 1 Report
An excellent publication providing great insight into a valuable approach to conducting research with children. Presentation of information empowers readers to conduct similar research within their own respective fields. Minor comments below relating to suggested changes, majority reinforcing with positive feedback! Very well done.
Abstract: great overview. However, consider rewording of the sentence at line 22 for easy of flow of reading.
Line 43: consider that not all authors will know what UNCRC stands for. Suggestion to use an abbreviation for first use - as seen at line 45.
Line 54: excellent! What a great sentence to summarise the benefits of the participatory research approach.
Introduction: great overview and inclusion of background information, identifying the gap in the current literature and the aims of the current study.
Line 79: to add greater context suggestion to add the age range of the youth advisory panel members.
Materials and method: excellent presentation of processes followed. Suggestion to include greater data relating to student demographics to give greater context to the findings.
Results: great outline of key findings and limitations identified by participants.
Discussion: excellent presentation of key points, practical considerations for translation and strengths and limitations.
Conclusion: great summary.
Author Response
Dear Reviewer
Many thanks for taking the time to review our manuscript. We appreciate your comments and suggestions and our responses are detailed below.
- Abstract: great overview. However, consider rewording of the sentence at line 22 for easy of flow of reading.
- We have revised this sentence and split in two to read: “The findings were found to be authentic, according to students, with the exception of two limitations. Students believed body image was not adequately represented in the findings, and alternative focus group compositions could have influenced discussions.”
- Line 43: consider that not all authors will know what UNCRC stands for. Suggestion to use an abbreviation for first use - as seen at line 45.
- This has been done
- Line 54: excellent! What a great sentence to summarise the benefits of the participatory research approach.
- Thank you
- Introduction: great overview and inclusion of background information, identifying the gap in the current literature and the aims of the current study.
- Thank you
- Line 79: to add greater context suggestion to add the age range of the youth advisory panel members.
- Age range 15-17 years added in now line 150.
- Materials and method: excellent presentation of processes followed. Suggestion to include greater data relating to student demographics to give greater context to the findings.
- Thank you for your suggestion. Unfortunately, we do not have additional demographic data for the students beyond age and their school characteristics which are detailed in Table 1.
Reviewer 2 Report
The paper describes an example of best practice in terms of participatory research approaches, with the innovation largely coming from the fact that this work was conducted in an adolescent population, where these approaches are under-explored. The inclusion of peer researchers within this setting is novel and has produced interesting findings and insights.
There does need a careful read through for clarity and structure as I do not believe the methodology is structured in the best order for clarity at present. Initially, I was not clear about use of the term peer researchers - at times "student pairs" are used or "students" and I believe these are the peer researchers but the language can be confusing. An early definition of peer researchers and the use of them would help, as the first mention is advice on the recruitment process. Some of this comes later, but the order needs to be considered - perhaps some of the quantitative study design, school and student recruitment should come before the participatory methodology as it would make much more sense with the order reversed.
Furthermore, I think a description of the purpose of the overall study (with the quantitative elements) would also be helpful earlier; as would a description of what the peer researchers were required to do (PhotoVoice described but photos of what - overall school environment or food environment?) - this may seem obvious to the authors but I think this context and order change would really help the independent reader.
The observation about a lack of discussion around body image is interesting, and it is observed that this may have been due to reticence among the participants. That is certainly true, but is it possible that the topic guide was constructed so as not to naturally lead to these sorts of observations/discussions? Or that the peer reviewers did not prompt and pursue this topic, even if it was mentioned by participants? That raises another question of whether experienced researchers as facilitators might have elicited different discussion/outcomes, and perhaps a combination of approaches might be useful to learn as much as possible from these qualitative methods? There is little doubt that inclusion of peer researchers is of value, but some further reflections as to the utility of this and whether there are limitations, and how they could be overcome, would be of interest.
Author Response
Dear Reviewer
Many thanks for taking the time to review our paper. We appreciate your comments and recommendations for improvements. Our responses to your points are outlined below in bold/italics.
There does need a careful read through for clarity and structure as I do not believe the methodology is structured in the best order for clarity at present. Initially, I was not clear about use of the term peer researchers - at times "student pairs" are used or "students" and I believe these are the peer researchers but the language can be confusing. An early definition of peer researchers and the use of them would help, as the first mention is advice on the recruitment process. Some of this comes later, but the order needs to be considered - perhaps some of the quantitative study design, school and student recruitment should come before the participatory methodology as it would make much more sense with the order reversed.
- Response: We have reviewed the methodology structure, and as you suggested opened with an overview now titled ‘study design’, continued with recruitment, then moved to quantitative methodologies and finally the participatory and qualitative elements. The term peer researcher is explained and language around this edited for consistency.
Furthermore, I think a description of the purpose of the overall study (with the quantitative elements) would also be helpful earlier; as would a description of what the peer researchers were required to do (PhotoVoice described but photos of what - overall school environment or food environment?) - this may seem obvious to the authors but I think this context and order change would really help the independent reader.
- Response: An introduction to the study has been added as the first section of the methodology (study design) which includes the overall aim and references existing quantitative and multi-stakeholder qualitative publications (Line 93-100). Instructions provided to students around food environment photovoice methodologies have been made clearer (See line 171-174).
The observation about a lack of discussion around body image is interesting, and it is observed that this may have been due to reticence among the participants. That is certainly true, but is it possible that the topic guide was constructed so as not to naturally lead to these sorts of observations/discussions? Or that the peer reviewers did not prompt and pursue this topic, even if it was mentioned by participants? That raises another question of whether experienced researchers as facilitators might have elicited different discussion/outcomes, and perhaps a combination of approaches might be useful to learn as much as possible from these qualitative methods? There is little doubt that inclusion of peer researchers is of value, but some further reflections as to the utility of this and whether there are limitations, and how they could be overcome, would be of interest.
- Response: These are interesting observations on this finding, thank you. We have expanded on this section of the discussion please see lines 371-386.
Reviewer 3 Report
This paper presents insightful methodological findings, and adds to the field in a number of ways. I think the paper got stronger, with a really strong discussion and conclusion presented. I do, however, feel that the background/introduction needs further thought (see below) and I have some smaller questions / suggestions for edits throughout.
- Background: Why are nutritional initiatives needed in the school setting? I think the paper would benefit from setting the scene first - for example, what do we know about prevalence of different dietary behaviours in adolescence?
- Background: I felt that the background made several sweeping statements / assumptions about the literature base. Most notably, there is a growing body of literature that focuses on socio-ecological factors that influence young people's food behaviours - the first paragraph glosses over this, and it would be good to see this work cited and fleshed out. Similarly, a growing body of literature does focus on participatory approaches with young people rather than children. A lot of the references in these areas are now quite dated and could be brought up to date? It would be good for the authors to proof read their reference list as one or two are incomplete.
- Methods: Did the authors use any INVOLVE principles or any of the principles of co-production in this work? If not, it may be good to acknowledge these somewhere, perhaps in the discussion?
- Methods: The rationale presented for the age range of participants suggests convenience? Is there any other reasons for focusing on 15-17 year olds?
- Methods: It would be good to see a reference to support the rationale used for peer-led focus groups?
- Methods: Analysis of focus groups is described as content analysis but it sounds like thematic analysis was conducted - could the authors clarify this?
- Findings: Could the age of respondents be presented to contextualise quotations?
Author Response
Dear Reviewer
Thank you for taking the time to review our paper. We appreciated your comments and recommendations. Our responses to your individual points are listed below in bold/italics.
- Background: Why are nutritional initiatives needed in the school setting? I think the paper would benefit from setting the scene first - for example, what do we know about prevalence of different dietary behaviours in adolescence?
- We have expanded the background to include more context. Please see lines 35-57.
- Background: I felt that the background made several sweeping statements / assumptions about the literature base. Most notably, there is a growing body of literature that focuses on socio-ecological factors that influence young people's food behaviours - the first paragraph glosses over this, and it would be good to see this work cited and fleshed out. Similarly, a growing body of literature does focus on participatory approaches with young people rather than children. A lot of the references in these areas are now quite dated and could be brought up to date? It would be good for the authors to proof read their reference list as one or two are incomplete.
- The socio-ecological background has been expanded, please see lines 44-57.
- The participatory elements have been updated as recommended please see lines 72-79.
- Methods: Did the authors use any INVOLVE principles or any of the principles of co-production in this work? If not, it may be good to acknowledge these somewhere, perhaps in the discussion?
- We did not use INVOLVE principles, but rather focused on youth orientated frameworks to design the collaboration phases. These were referenced in the background, methodologies and now more fully in the discussion, please see lines 326-334.
- Methods: The rationale presented for the age range of participants suggests convenience? Is there any other reasons for focusing on 15-17 year olds?
- The other reason was senior school cycle, with greater autonomy to leave the school campus at break times. This has been made clearer in the methods, please see lines 106-109.
- Methods: It would be good to see a reference to support the rationale used for peer-led focus groups?
- We have edited the methods section to make our rationale for choosing peer led focus groups clearer, please see section 2.6.2. lines 189-198. There are surprisingly few published studies evaluating peer led focus groups (we have now added this point to the discussion, please see line 301-309 for a comparison with a new citation on peer led focus groups published this year).
- Methods: Analysis of focus groups is described as content analysis but it sounds like thematic analysis was conducted - could the authors clarify this?
-
- It was content analysis according to the reference we cite (Hsieh & Shannon, 2005). Given the multi-methods outlined in the current paper, it is challenging to provide additional depth within space available. Additional detail on the analysis process is explained in our paper: Browne S, Barron C, Staines A, Sweeney MR. 'We know what we should eat but we don't ...': a qualitative study in Irish secondary schools. Health Promot Int. 2020;35(5):984–993
- Findings: Could the age of respondents be presented to contextualise quotations?
- Unfortunately, at this stage it is not possible to retrospectively retrieve ages of participants, who were all 15-17 years. Pseudonyms were applied at the transcription stage and exact ages of each participant were not assigned.